# Is It Possible to Educate, Intervene or “Cure” Autism Spectrum Disorder? A Content Analysis of YouTube Videos

**DOI:** 10.3390/ijerph18052350

**Published:** 2021-02-28

**Authors:** Irene Lacruz-Pérez, Pilar Sanz-Cervera, Gemma Pastor-Cerezuela, Irene Gómez-Marí, Raúl Tárraga-Mínguez

**Affiliations:** 1Department of Education and School Management, Faculty of Teacher Training, University of Valencia, 46022 Valencia, Spain; pilar.sanz-cervera@uv.es (P.S.-C.); irene.gomez@uv.es (I.G.-M.); raul.tarraga@uv.es (R.T.-M.); 2Basic Psychology Department, Faculty of Psychology, University of Valencia, 46010 Valencia, Spain; gemma.pastor@uv.es

**Keywords:** autism spectrum disorder, content analysis, internet, intervention, social media, YouTube

## Abstract

YouTube is one of the most well-known and widely accessed websites worldwide, thus having a powerful pedagogical potential. Nonetheless, the quality and the veracity of some YouTube videos are questionable. Doubts regarding the trustworthiness of factual content is a controversial factor that needs to be taken into account, especially when addressing public health issues. For this reason, the main objective of this work is to analyze the content of the most viewed videos in Spanish on YouTube related to autism spectrum disorder (ASD). To carry out this research, the terms “autism AND education”, “autism AND intervention”, and “autism AND cure” were used (in Spanish). The analysis of these searches results indicated that videos included in the “cure” category are shorter, and less valued by internet users, obtaining the lowest ratings on the “Patient Education Materials Assessment Tool” for audiovisual materials (PEMAT-A/V), in addition to present therapies that are in fact more harmful than videos in other categories. In general, videos containing recommendations for therapies that are harmful are the ones that have received most views, along with the videos that include alternative non-harmful therapies. Practical implications of these findings and recommendations for further research are discussed.

## 1. Introduction

The use of the internet as a source of information has increased substantially in recent years. One of the most well-known and widely accessed websites worldwide is YouTube. This video-sharing site has two different functions, since it is not only a video repository, but also a social network interface in which users can share and rate videos, and comment on them [1].

Nowadays, YouTube is increasingly being used as a platform for disseminating health and education information [2,3]. With regard to the field of special education, autism spectrum disorder (ASD) is one of the diagnoses most addressed on YouTube [4], which cannot be surprising given its increasing prevalence [5]. ASD is a heterogeneous neurodevelopmental disorder characterized by the presence of persistent difficulties in communication, social interaction, and the existence of restrictive and repetitive patterns of behavior, interests, or activities [6].

The videos on YouTube that deal with this disorder often use children, youths, and adults with autism as the protagonists [1,7], as well as the families of children with this diagnosis, all of whom narrate their experiences from their own perspectives [8,9]. Several organizations, specialized in autism, upload videos on this site, in which experts such as teachers, pediatricians, or psychologists, discuss fundraising events, interventions, and therapies, among other information [10].

Searching for this type of video on YouTube to gather data about autism is not complicated and results in multiple benefits [4]. First, finding information from other families and people living with or who have ASD may help the researchers feel less isolated, as they can share experiences, as well as help others to understand what it is like to grow up and live with the disorder [10]. Second, YouTube videos can facilitate detection of core features of ASD [8,11], allowing for early identification. Third, YouTube videos from governmental organizations and professional associations, which contain trustworthy and high-quality data about diagnosis and treatment resources, can be used as powerful educational tools [3]. In addition, people with ASD can create videos by themselves, without the overlay of direct social interaction with others, making their own needs and interests known worldwide [4], and thus giving them a voice to educate others not only about autism itself, but also about how they think and feel [10]. In this sense, YouTube videos could promote a move to more social inclusion of people with this disorder [7,9,12].

Despite these benefits and being that it is one of the websites in which users have greater freedom of speech [13], viewing videos on YouTube as a source of medical information can also have drawbacks since there are minimal guidelines regulating the content of the materials uploaded. Additionally, it is a public platform, free of charge, where users need to do no more than create an account to be able to post videos. These requirements lead professionals and researchers to question the veracity and quality of the information found on the platform. In fact, previous studies have found that many YouTube videos contain misleading information, primarily anecdotal, which contradict the reference standards [3,12]. In this sense, some studies conclude that this platform can be used as a medium for promoting unscientific therapies, as well as having the power to influence people’s beliefs, one way or another, on controversial topics [3,8]. One such example is the misconception of linking vaccines with the onset of autism [13].

Researchers have found that videos with the most views have been uploaded by non-professionals and provide limited content [12,14]. Although there are many that are educationally useful, such as those created by the families of children with autism [8,9], other videos have been found to contain erroneous information, such as, for example, those that demonstrate parental implementation of specific teaching strategies of the picture exchange communication system (PECS) [15].

Given the popularity of YouTube, and considering its advantages and disadvantages, a content analysis of the videos uploaded to the site that are related to autism, is indicated. Even though research regarding this subject has increased in recent years, none has focused on videos presented in Spanish. For this reason, the main objective of this work is to analyze and to evaluate the content of the most viewed videos in Spanish addressing ASD uploaded onto YouTube. This objective is innovative as there is special relevance to favor the social inclusion of people with ASD and their families.

According to previous studies, there are at least two aspects that need to be taken into account when doing a content analysis on YouTube videos: first, video-related data, such as number of views, length, and number of likes (thumbs up) and dislikes (thumbs down), are features that may be related to the quality of the videos [14]; second, multiple-word search is a factor that can procure more specific results and larger variability over time [16].

The factors included in high-density searches, then, influence the focus of the two main objectives for this study:To analyze the most viewed YouTube videos in Spanish, identified when searching the following terms in Spanish: “autism AND education” (“autismo AND educación”), “autism AND intervention” (“autismo AND intervención”), and “autism AND cure” (“autismo AND curación”). There are three types of variables in which these three blocks of videos are compared: (a) the metadata collected on YouTube, including the duration of the videos, the number of views, the number of thumbs up and the number of thumbs down; (b) its understandability and actionability evaluated through the “Patient Education Materials Assessment Tool for Audiovisual Materials” (PEMAT-A/V) [17,18]; and (c) the type of content included.To analyze the number of views, the number of thumbs up, and the number of thumbs down of the most viewed YouTube videos in Spanish whose content invites the viewer to follow harmful therapies or interventions.

These objectives are considered exploratory since to date we have not found any other studies that analyze the most viewed YouTube videos about ASD in Spanish. In addition, we have not found previous studies that perform a multiple-word search when analyzing the most viewed YouTube videos about ASD. This is a relevant aspect since it can provide information having to do with the search terms that identify videos with more scientific rigor and so greater importance for internet users looking for reliable information on the topic. They also help to identify the search terms that include more potentially harmful or dangerous content. Additionally, the second objective may be useful to gauge the impact on users of videos that propose harmful and/or dangerous content for people with ASD and their families.

Considering previous pioneering studies that have analyzed the most viewed YouTube videos about ASD in English, we hypothesize that most of the content will be related to signs and symptoms [14], promoting an early detection of the disorder [11,12]. Regarding the understandability and actionability that analyze the PEMAT (A/V), we hypothesize that videos included in the “cure” category will be less valued by internet users than the other two types of videos, being that previous studies reveal that families of children with ASD lean toward watching videos made by professionals [14]. As for the analysis of the most viewed YouTube videos whose content invites the viewer to follow harmful therapies or interventions, we have not found either any other study that analyze this aspect. Nonetheless, some research shows that support for a link between vaccines and autism is most prominent on YouTube [13], so it is possible that videos which recommend harmful therapies have a lot of views, being a danger for the optimal development of people with ASD.

## 2. Materials and Methods

### 2.1. Videos Selection

To select the videos to be analyzed, three searches were made on YouTube with the terms (in Spanish): “autism AND cure” (“autismo AND curación”), “autism AND intervention” (“autismo AND intervención”), and “autism AND education” (“autismo AND educación”). The results of the different searches were ordered by number of user views. The first 50 videos from each of the searches were selected, of which a total of 150 videos were analyzed. Six of the videos initially selected were eliminated for different reasons: one video was eliminated because it was narrated in Portuguese (the work focuses on videos presented in Spanish). Another was removed for its duplication in the same search with two different URLs (this video was, in the end, counted only once). Another one was removed because it contained a technical problem which muted the audio. Another one was removed for having only an audio track that repeated apparently nonsensical phrases with no relation to ASD. Finally, two other videos were removed because they appeared simultaneously in two of the searches carried out. These six deleted videos were replaced by the next videos in the list of results with the highest number of views.

### 2.2. Procedure

YouTube searches were carried out during the last week of October 2020. For the content analysis, a series of guidelines were prepared according to the indications in Section 2.3. Two of the authors of this study conducted the analysis. Both of them are part of a research group that has conducted similar studies to this one, and they work as a research staff and professors in a Teacher Training Faculty of a public university in Spain.

Both authors independently viewed five videos, analyzed their content, and carried out the coding in independent databases. The results were then compared in order to clarify issues that were not clear in the initial guidelines. Once these guidelines were refined, the two authors independently analyzed and coded 90 videos, so that 30 of the 150 videos were analyzed by the two authors. Subsequently, the Cohen’s kappa coefficient of concordance was calculated for the results of the 30 videos that were double encoded, finding a result of K = 0.89, which corresponds to the “almost perfect” range, according to [19]. The discrepancies found between the evaluations of the two authors were resolved by consensus.

The data extracted from the 150 videos analyzed are available in Appendix A.

### 2.3. Variables Analyzed from the Videos

For each video, three types of variables were analyzed:(a)Metadata: the number of views, duration, and number of likes and dislikes indicated by the users of the platform were recorded.(b)“The Patient Education Materials Assessment Tool for Audiovisual Materials” (PEMAT-A/V) [17,18]. It is a questionnaire that evaluates patient education materials. It includes 17 items with two answer options: agree and disagree. In eight of the items there is also a third option of “not applicable”. The questionnaire has two scales, described by its authors, as follows: (1) Understandability (13 items): patient education materials are understandable when consumers of diverse backgrounds and varying levels of health literacy can process and explain key messages; and (2) Actionability (4 items): patient education materials are actionable when consumers of diverse backgrounds and varying levels of health literacy can identify what they can do based on the information presented.(c)Content analysis: the content of each video was analyzed, classified in one of these categories:
Conventional interventions with proven effectiveness: this category includes videos whose content proposes one or more of the following options: pharmacological interventions prescribed by health professionals after carrying out clinical evaluations of the patient with ASD; psychoeducational interventions based on Applied Behavior Analysis (ABA); interventions based on Treatment and Education of Autistic and Related Communication Handicapped Children (TEACCH); recommendations to carry out interventions by a professional at the earliest possible moment; interventions to improve language, communication skills, or social interaction skills carried out by licensed professionals; and training family members in behavioral management and structuring of the family environment.Harmful or dangerous interventions: this section includes videos that propose interventions that pose risks to the health of children with ASD (use of substances such as sodium chlorite, cannabis, or extreme diets that can endanger their health); videos that, despite proposing innocuous treatments, encourage families to abandon any other type of intervention outside the one proposed; and those in which results are promised in a very short period of time, so they can be described as miraculous, since in practice it would be extremely unlikely to achieve these results.Denying harmful or dangerous treatment: this category includes videos whose main objective is to deny the effectiveness of harmful interventions for children with ASD and/or to denounce their danger.Alternative interventions that can complement conventional interventions: this classification includes videos in which non-harmful intervention procedures are proposed, learning towards the positive regarding some aspect of the development of children with ASD, but which do not yet have robust empirical support to confirm that they are effective (for example: music therapy or art therapy). Videos that recommended complementary interventions to conventional ones were also included. This includes interventions which can be generally positive for the health of any individual, but which do not entail specific benefits for ASD symptoms, such as diets based on varied and healthy structures, an active and healthy lifestyle or carrying out activities related to art or creativity. Videos that exclusively proposed one or more of these procedures or recommended abandoning other treatments proposed by health or education professionals were not included in this category.Information about ASD: this section includes videos that provide basic information about one or more of the following aspects related to ASD: diagnostic criteria, its prevalence, evaluation procedures, or implications for the family. These videos did not propose specific intervention procedures.Right to inclusive education for people diagnosed with ASD: this category includes videos that deal with issues related to the rights of students with ASD to have an inclusive education in mainstream schools, and the provision of specialists, materials, and organizational resources necessary to achieve the children’s maximum development potential.

### 2.4. Statistical Analysis

After verifying that the data did not meet the assumption of normality, we opted to use non-parametric statistical tests. For the first objective, in order to analyze whether there were statistically significant differences among the videos located from the three different search terms (“autism AND education”; “autism AND intervention”; “autism AND cure”), in the duration of the videos, the number of views, the number of thumbs up and down, and the PEMAT (A/V) Understandability and Actionability scores, Kruskal-Wallis tests for independent samples and pairwise comparisons were performed. The values of significance were adjusted by the Bonferroni correction for multiple testing. Second, to analyze the distribution of the type of content in the three searches, a chi square statistic was used. To carry out the second objective, with the aim of analyzing the number of views, the number of thumbs up and thumbs down of the most viewed YouTube videos whose content suggests following harmful therapies or interventions, new Kruskal–Wallis tests for independent samples and pairwise comparisons were performed, adjusting the values of significance by the Bonferroni correction for multiple testing.

## 3. Results

### 3.1. Analysis of the Most Viewed YouTube Videos in Spanish, Identified When Searching the Terms “Autism and Education”, “Autism and Intervention”, and “Autism and Cure”

The results of the Kruskal–Wallis tests, performed to analyze whether there were differences in terms of the duration of the videos, the number of views, the number of thumbs up, the number of thumbs down, and the Understandability and Actionability scores of the PEMAT (A/V) amongst the three groups of videos located from the different search terms, determined the existence of statistically significant differences in the duration of the videos, the number of thumbs down given by YouTube platform users’, and the Understandability and Actionability scores of the PEMAT (A/V). In contrast, there were no statistically significant differences in the number of views, or the number of thumbs up given by YouTube platform users’ (Table 1).

The pairwise comparisons showed that the videos obtained with the search terms “autism AND cure” are shorter in duration than those obtained with the search terms “autism AND intervention”, and they also received a greater number of thumbs down than the videos obtained from the other searches. Regarding the Understandability and Actionability scales of the PEMAT (A/V), the pairwise comparisons showed that the videos located in the search “autism AND cure” obtained lower ratings on the Actionability scale than those obtained from the search with the keywords “autism AND education”, and lower ratings on the Understandability scale than the videos obtained from the other two searches.

To answer the question, “What content do users find when they perform the different searches?” a contingency table was made (Table 2). This table includes an analysis of the distribution considering the type of content based on the searches carried out. The chi square statistic shows that this distribution is not homogeneous *χ*^2^ = 112.207_(10),_
*p* ≤ 0.001; Cramér’s V = 0.612. Table 2 demonstrates that all videos recommending harmful therapies were obtained from the search “autism AND cure”. In addition, 20 of the 24 videos whose content focuses on disproving harmful therapies were also obtained in this same search.

### 3.2. Analysis of the Number of Views, the Number of Thumbs up, and the Number of Thumbs down of the Most Viewed YouTube Videos Whose Content Invites Us to Follow Harmful Therapies or Interventions

After verifying the high presence of videos related to dangerous content (8.67% of the total videos analyzed) and given the interest in analyzing the real extent of the impact on users of these videos due to their risk to public health, the authors decided to compare the videos of each content category and the information regarding the metadata. Specifically, the number of views, the number of thumbs up and the number of thumbs down of the videos of the different thematic categories were compared.

To do this, new Kruskal–Wallis tests for independent samples were carried out. The aim was to compare these variables (the number of views, the number of thumbs up and the number of thumbs down) between the six types of video content: (1) videos that explain and/or exemplify conventional interventions in ASD with effectiveness supported by research (*n* = 77); (2) videos that deny harmful or dangerous treatments (*n* = 24); (3), videos that explain how to carry out non-harmful alternative interventions (although without reliable support by research) (*n* = 15); (4) videos with generic information about ASD that do not explain in detail a specific intervention procedure (*n* = 14); (5) videos explaining harmful or dangerous interventions for people with ASD and/or their families (*n* = 13); and (6) videos that focus on aspects related to the right to an inclusive and quality education for people with ASD (*n* = 9).

The Kruskal–Wallis tests showed that differences between the videos grouped by content type turned out to be statistically significant in the three analyzed metadata variables: the number of views, the number of thumbs up and the number of thumbs down.

The pairwise comparisons, included in Table 3, show that the videos that explain how to carry out non-harmful alternative interventions had more views than those that claim to deny the efficacy of harmful therapies, and those that provide information on ASD. In addition, videos that provide information on ASD received a lower number of thumbs down than videos recommending harmful therapies, videos claiming to disprove the efficacy of harmful therapies, and videos explaining how to carry out non-harmful alternative interventions. Moreover, those videos that claim to deny the efficacy of harmful therapies received more thumbs down than those addressing the right to inclusive education. In the case of the pairwise comparisons on the number of thumbs up between the six types of video content, videos recommending harmful therapies received more thumbs up than those that claim to deny the efficacy of harmful therapies, although this difference was marginally significant (*p* = 0.07).

## 4. Discussion

### 4.1. Discussion of Results

In line with the initial hypothesis, results indicate that videos included in the “cure” category are less valued by internet users, obtaining lower ratings on the Actionability and Understandability scales of the PEMAT (A/V) than the other two types of videos. Additionally, the “cure” videos are shorter than the videos included in the “intervention” category, and they are more related to harmful therapies than the other videos. In fact, all videos recommending harmful therapies were obtained with the search “autism AND cure”, and none were obtained by using the terms “education” or “intervention”. Fortunately, searches in YouTube using the term “cure” were also responsible for the identification of 20 out of 24 videos whose main content was related to denouncing the dangers of these harmful therapies and/or denying their supposed effectiveness.

Although the above-mentioned results are positive, as those videos can be a counterpoint to the ones that propose dangerous therapies, they in fact create a veil of deception, as they give the impression that there is a legitimate debate between defenders of one position or another. For this reason, professionals from educational and health fields should emphasize that autism is a condition and not a disease that can be cured. Previous research has found that when searching “curing autism” on YouTube, 2500 results were obtained, most of them with higher views than “living with ASD” videos [9]. As the authors of this study want to emphasize, videos that purport cures may initiate false hope in parents, preventing them from accepting and coping with the diagnosis. For this reason, it is indicative to use vocabulary in deliberate ways. The conscious use of vocabulary and the awareness that the term “cure” in relation to ASD ought to be rejected, may depend on the behavior of internet searches on YouTube.

Such rejection of deceptive messages is difficult considering the fact that previous studies have found very few videos on YouTube uploaded by professionals and have instead determined that the most frequently viewed videos are personal and television-based [12]. Nonetheless, consideration needs to be given to previous studies which concluded that the mean number of views of videos uploaded by professionals was notably higher than consumer and internet-based videos [14].

The source of the video is not necessarily an indication of their quality. In fact, previous research confirm that personal videos of families of children with ASD [8,9], as well as videos in which first-person accounts of people with ASD appear [1,7,10], can provide high quality information. This is very valuable for professionals in the health and education fields, since the reality of what it means to live with the disorder can be understood to a greater extent, and relevant resources and interventions adapted to the needs of people with ASD (and/or their families) will be more accessible. Nevertheless, it is true that professionals should provide more quality content on platforms such as YouTube, taking into account the exponential impact that this platform has had in recent years [4,5]. Some studies have found several videos in which families make errors using PECS with their children in non-research conditions and without the support of an instructor [15]. These errors might be propagated, as other parents may use the videos as models, so it indicated that professionals must take action against the spread of such videos.

As [14] points out and it was hypothesized in this paper, the majority of YouTube videos present information that include signs and symptoms, and this is positive regarding detection, but do not facilitate knowledge about the treatment. Research suggests that online videos can promote early detection of ASD [11]. However, the usefulness of the treatment available that is presented in these videos is controversial [12]. Beyond early detection, some research has found that parents use the internet to aid them in making decisions about intervention after receiving a diagnosis of ASD [20].

The conclusions are then that there is a high presence of videos that include content that encourages harmful therapies or interventions. Additionally, meeting one of the hypotheses of this research, this type of videos almost receives the greatest number of views, since they are just preceded by the videos that display alternative interventions that are non-harmful, but they do not have scientific endorsement either.

This is an aspect of public health that needs to be studied in depth, since its impact can cause serious challenges with regard to the health of people with ASD. In addition to the fact that videos promoting harmful therapies have large number of views, a marginally statistically significant difference has been found between the number of likes of videos including dangerous treatments in their content and the number of likes of the videos that precisely deny harmful treatments. This result is in accordance with the review made by [21], who found that videos with misinformation about ASD tend to have high levels of popularity among users. Previous studies have already suggested the high degree of influence that YouTube videos containing misinformation about ASD can have on internet users [22]. These findings are highly dangerous, as they can lead to many misconceptions, as well as spreading myths, that create an environment in which the social inclusion of people with ASD becomes difficult.

### 4.2. Limitations and Future Research Directions

This study includes several limitations. First, it has only focused on one popular video-sharing site, without considering videos included in other popular social media platforms. Second, as [2] claims, it is worth considering that the quality of information on YouTube is unclear and not standardized. Lastly, videos included in this study were only those presented in Spanish. This is one of the strengths of this work, since it is the only study that analyzes the content of YouTube videos in Spanish so far; however, it is also a limitation since the results obtained are not generalizable to other languages.

Future studies should analyze the videos included in other popular social media platforms, as well as the content of videos in different languages in the same study. The fact of analyzing the videos comments’ is also an aspect that could be considered in future works, since this analysis can provide a significant amount of information about the beliefs that the population has about the disorder. Moreover, future researchers could invite more experts to evaluate the videos.

## 5. Conclusions

YouTube videos may be useful for disability awareness as well as early detection and intervention by prompting caregivers to seek guidance. Personal stories also hold powerful educational value for professionals who are attempting to understand the daily life of people with ASD and their families, with the aim of providing effective interventions. It can also be a platform that promotes the social inclusion of the population with any type of special educational needs. Nonetheless, the usefulness and the veracity of the available videos are questionable. For this reason, it is important to make a critical use of the information, considering the source of the videos. Given the wide reach of social media, an increased professional presence is imperative in order to provide accurate information about ASD. Videos created and uploaded by professionals could also address the safety and efficacy of treatments, especially the ones that are harmful. These videos could also include some links to credible sources for additional information and available resources on the net. In this way, the families of children with ASD and the professionals who work with them, as well as the entire population, could have more quality information about this disorder, thus favoring the necessary social inclusion of people with this diagnosis.

## Figures and Tables

**Table 1 ijerph-18-02350-t001:** Average ranges and Kruskal–Wallis H-test statistic values for all three YouTube searches metadata.

Search Terms/Metadata	Autism AND Education (EDU) (*n* = 50)	Autism AND Intervention (INT) (*n* = 50)	Autism AND Cure (CUR) (*n* = 50)	Kruskal–Wallis H	*p*	η^2^_H_	Group Differences
	AR	AR	AR				
Number of views	81.23	68.06	77.21	2.414	0.299	0.003	-
Video length	75.22	89.95	61.33	10.887 **	0.004	0.060	INT > CUR
Thumbs up	80.67	68.39	77.44	2.147	0.342	0.001	-
Thumbs down	66.14	61.97	98.39	21.226 **	<0.001	0.131	CUR > EDU, INT
Understandability (PEMAT-V)	86.49	86.99	53.02	21.425 **	<0.001	0.132	EDU, INT > CUR
Actionability (PEMAT-V)	86.56	78.65	61.29	9.614 **	0.008	0.052	EDU > CUR

** *p* < 0.01.

**Table 2 ijerph-18-02350-t002:** Table of content contingencies according to search terms.

Type of Content/Search Terms	Conventional Interventions	Denying Harmful or Dangerous Treatment	Non-Harmful Alternative Interventions	Information about ASD	Harmful Therapies	Right to Inclusive Education
Autism AND education (EDU) (*n* = 50)	22 (44%)	2 (4%)	7 (14%)	11 (22%)	0 (0%)	8 (16%)
Autism AND intervention (INT) (*n* = 50)	45 (90%)	2 (4%)	2 (4%)	2 (4%)	0 (0%)	1 (2%)
Autism AND cure (CUR) (*n* = 50)	10 (20%)	20 (40%)	6 (12%)	1 (2%)	13 (26%)	0 (0%)
Total	77 (51.33%)	24 (16%)	15 (10%)	14 (9.33%)	13 (8.67%)	9 (6%)

**Table 3 ijerph-18-02350-t003:** Average ranges and Kruskal–Wallis H-test statistic values for metadata based on content type.

Search Terms/Metadata	Conventional Interventions (CON) (*n* = 77)	Denying Harmful or Dangerous Treatments (DEN) (*n* = 22)	Non-Harmful Alternative Interventions (ALT) (*n* = 15)	Information about ASD (INF) (*n* = 14)	Harmful Therapies (HARM) (*n* = 13)	Right to Inclusive Education (EDU) (*n* = 9)	Kruskal– Wallis H	*p*	η^2^_H_	Group Differences
	AR	AR	AR	AR	AR	AR				
Number of views	76.25	55.64	102.50	51.43	96.15	80.22	17.758 **	0.003	0.089	ALT > DEN, INF
Thumbs up	75.52	57.95	94.97	72.79	100.88	53.33	13.436 *	0.020	0.059	HARM > DEN
Thumbs down	71.14	100.02	90.70	40.07	99.35	48.17	26.630 **	<0.001	0.150	HARM, DEN, ALT > INF; DEN > EDU

* *p* < 0.05; ** *p* < 0.01.

## Data Availability

The data presented in this study are available in Appendix A.

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
