# Peer review of "Is It Possible to Educate, Intervene or “Cure” Autism Spectrum Disorder? A Content Analysis of YouTube Videos"

_ijerph, 2021, doi:10.3390/ijerph18052350_

Round 1

Reviewer 1 Report

In this manuscript, the authors analyzed the content of the most viewed videos on YouTube related to ASD, and analyzed those. 

I have found too many drawbacks in this manuscript, starting from its idea, through the design, and of course, the potential interpretation. 

I can not see how "number of views" reflects something very much specific. Therefore, I can not understand why count on it, or discuss anything related to this aspect. 

The reason for pressing "like" or "dislike" is not necessarily known to the authors, and can not be used as a measure for something. 

The manuscript relies on too many factors that their interpretation or meaning are doubtful. This makes the meaning of the manuscript in doubt.

What is the meaning of the findings related to the "cure" category? How can this manuscript improve our understanding of ASD or its "cure"?

This manuscript might be perhaps relevant to other journals, perhaps media-related, but I don't see it suitable for IJERPH, since I don't find this manuscript of high quality, or of interest to the broad audience of this journal. 

Reviewer 2 Report

The article concerns a very important and current issue, which is the reaction of people to videos posted on the popular social media channel, which is YouTube. The authors focused on Spanish-language films on autism spectrum disorders. The article is well written and prepared. All the work is consistent and you can see a smooth transition from one point to another. The topic is topical and interesting. The topic is not new, but the research done should be considered as the authors' contribution.
The aim of the article is to analyze the content of YouTube videos in Spanish related to autism spectrum disorders. However, the goal is also to evaluate these videos, not just to analyze. Therefore, it would be worth developing the adopted goal.
The literature review (included in the introduction) is quite good, although it is worth considering some issues relating to autism. The description focuses solely on the characteristics of the medium which is YouTube. Quite current literature was used. Despite the fact that it is a research article, the authors could consider separating the "Literature Review" section.
The authors know the limitations that create opportunities for future research. However, it is worth considering separating the section on limitations and future research directions from the "Discussion" section.
There are no hypotheses or even research questions in the article. This is a research paper, so it would be worthwhile to supplement it with it.
The research methodology and the use of analytical methods and tools are correct. More experts could be invited to evaluate the films in terms of the adopted criteria. This may be the direction of further research. The authors conducted statistical analyzes.
The discussion section is well structured and the conclusions may be of interest to the reader. The article matches the publication profile. The results were interpreted accordingly.
The article is worth reviewing for the correct English language as it contains some grammatical errors.

Reviewer 3 Report

The authors sought to evaluate videos on YouTube in Spanish for the cure, intervention, and education of autism. They chose the first 50 videos with most views in each category at the time of the study. Six inappropriate videos were discarded to be replaced with the next ones viewed most frequently.

The authors sought to evaluate the classes of videos for number of views, thumbs up ratings, and thumbs down ratings; understandability: and content.

Two authors independently rated the videos. What are the credentials of the authors who performed the ratings?

The authors then sought to perform parametric statistical procedures to examine relationships among items. The normality of the underlying distributions is an assumption for parametric analyses. Did the authors demonstrate the normality of populations by the Shapiro-Wilk or Kolmogorov-Smirnov Test? If the populations are not normally distributed, then a nonparametric test like Kruskall-Wallace may be used.

The authors conclude that some of the most viewed videos contain interventions that may be harmful.

The authors have identified an important problem, the use of videos to obtain information about diagnosis and intervention for autism in Spanish. Since large populations around the world speak Spanish, this is a widely used tool. 

The study will likely merit publication with revisions.

The manuscript will be enhanced by the tabulation of the exact YouTube sites included in each category. This could be constructed as a group of supplementary files identifying each video classified in each category by the authors. A flow chart to identify the specific videos will be useful for readers who seek to replicate the study. Also files of videos rated by understandability and content will be valuable. Since these files may exceed limits for supplementary files, they could could be published separately in repositories  such as Data, Zenodo, or Mendeley.

Page 2

Lines 93 and 94

In addition to providing the translations into English, state the original words in Spanish.

Page 3

Lines 113 and 114

In addition to providing the translations into English, state the original words in Spanish.

Round 2

Reviewer 1 Report

I thank the authors for the detailed responses and clarifications. I have no further issues with the manuscript. 

Author Response

Dear reviewer,

Thank you very much for contributing to improve the quality of our work.

Yours sincerely

Reviewer 3 Report

The authors seek to assess the quality and veracity of items about autism in Spanish on YouTube. They assessed commonly viewed videos in several reasonable categories. They then performed analyses to estimate the value of several categories of responses. Since there is a huge audience of people around the world seeking information in Spanish about autism, readers will benefit from objective measures of the quality and veracity of the items.

The manuscript will likely merit publication with revisions.

Line 255

Change “Ranges” to “ranges”

Line 257

In the legend add “AR:Average range”

Line 306

Change “Ranges” to “ranges”

Supplementary files

I am unable to figure out how to view the .sav file on my apple equipment. Readers may be unable to open files in the .sav format. The supplementary files could possibly be published in multiple formats to provide readers several options.  Please provide it in a pdf or excel format so that I can fully examine it.

Author Response

Dear Reviewer,

We have included the proposed changes:

  • In line 255 – we have changed “Ranges” to “ranges”
  • In line 257 – we have added to the legend “AR: Average range”
  • In line 306 – we have changed “Ranges” to “ranges”

In order to make it accessible to any interested reader of the article, we have added the supplementary file in .pdf format. (in addition to the .sav format).